# Impact of chronic diseases on the periapical health of endodontically treated teeth: A systematic review and meta-analysis

Bianca Marques de Mattos de Araujo[1,2]*, Bruna Marlene de Miranda[2,3], Tatiana Carvalho Kowaltschuk[1,2], Flávio Magno Gonçalves[2], Angela Graciela Deliga Schroder[2,3], Erika Calvano Kuchler[3], Odilon Guariza-Filho[2,4], Everdan Carneiro[1], Cristiano Miranda de Araujo[2,3], Ulisses Xavier da Silva-Neto[1]

1 Postgraduate Program in Dentistry, Department of Endodontics, Pontifícia Universidade Católica do Paraná, Curitiba, Paraná, Brazil, 2 Center for Advanced Studies in Systematic Review and Meta-Analysis – NARSM, Curitiba, Paraná, Brazil, 3 Postgraduate Program in Dentistry, Department of Endodontics, Tuiuti University of Paraná, Curitiba, Paraná, Brazil, 4 Postgraduate Program in Dentistry, Department of Orthodontics, Pontifícia Universidade Católica do Paraná, Curitiba, Paraná, Brazil

* mattosbianca@hotmail.com

**Data Availability Statement:** All relevant data are within the manuscript and its Supporting information files.

## Abstract

Systemic diseases affecting the immune system can influence the body's response time to endodontic treatment, potentially necessitating a longer duration for the complete resolution of existing infections when compared to healthy controls. This systematic review aims to evaluate the association between the presence of chronic diseases and periapical status after endodontic treatment through a systematic and comprehensive assessment of existing literature on this topic. The search strategy covered seven electronic databases and grey literature, encompassing articles published until October 2023. Two reviewers independently assessed potentially eligible studies based on the following criteria: Included were studies involving populations exposed to pre-existing chronic diseases who underwent endodontic treatment in permanent teeth. These studies evaluated periapical health status, making comparisons with healthy individuals. There were no language or publication date restrictions. Additionally, two reviewers independently extracted data regarding the characteristics of the included studies. The risk of bias was assessed using the Joanna Briggs Institute Critical Assessment Checklist. Meta-analysis was conducted using random effects models. The certainty of evidence was assessed using the GRADE tool. Twenty-three studies were included in the synthesis. Patients with diabetes were found to have about half the odds of having periapical health compared to non-diabetic patients (OR = 0.46; 95% CI = 0.30–0.70%; $I^2$ = 58%) in teeth that underwent endodontic treatment. On the other hand, other systemic diseases like HIV, cardiovascular disease, and rheumatoid arthritis did not demonstrate significant differences concerning the outcome. In conclusion, diabetic patients showed a lower likelihood of maintaining periapical health. Conversely, patients with HIV, cardiovascular disease, and rheumatoid arthritis did not exhibit significant differences, although the existing evidence is still considered limited. It is crucial to manage these patients in a multidisciplinary manner to provide appropriate care for this population.

**Funding:** The author(s) received no specific funding for this work.

**Competing interests:** The authors have declared that no competing interests exist.

## Introduction

Studies have highlighted the role of oral infections as contributing factors to the emergence of various systemic diseases [1, 2]. The process initiated by dental caries affects the dental structure, and when its progression occurs, the infection initially located in the dental pulp can spread to supporting structures, including bone tissue, thereby increasing the level of systemic exposure to these pathogens [3]. If endodontic treatment is not performed, the root canal becomes a source of predominantly gram-negative bacteria [4], potentially leading to systemic repercussions.

Endodontic treatment is aimed at addressing both pulpal and periodontal tissues, requiring systematic chemical and mechanical cleaning and the sealing of canals and dental crowns. All these measures are necessary to ensure a favorable prognosis, preventing contamination and reinfection [5]. However, it has been suggested that the systemic health condition may influence the outcome of endodontic treatment [6]. Systemic diseases affecting the immune system can influence the body's response time to endodontic treatment, possibly requiring a longer time for the complete resolution of existing infections when compared to healthy controls [7]. Patients with systemic diseases such as diabetes mellitus, hypertension, and coronary artery disease may have an increased risk of tooth extraction after non-surgical endodontic treatment [8].

A systematic review addressed the association between chronic diseases such as diabetes, cardiovascular disease, and the human immunodeficiency virus (HIV), and the outcomes of endodontic treatment. The results were inconclusive regarding the relationship between cardiovascular disease and diabetes with endodontic treatment outcomes. Additionally, no significant association was found between HIV-positive patients and endodontic treatment outcomes [9]. However, it is important to note that this review was limited to only three databases and restricted its search until the year 2016, which precluded the conduct of a quantitative analysis through meta-analysis. Other reviews have addressed specific chronic diseases, but none aimed to assess the impact of chronic diseases more comprehensively [10, 11]. Furthermore, it is worth noting that new relevant studies have been published on this topic. Therefore, the need for a new systematic review with a comprehensive search strategy that includes a larger number of databases, as well as grey literature, to provide a more comprehensive and up-to-date overview is justified. It is also relevant to investigate whether other chronic diseases may influence the repair process following the endodontic treatment.

Therefore, this systematic review aims to evaluate the association between the presence of chronic diseases and the periapical status after endodontic treatment through a systematic and comprehensive assessment of existing literature on this topic.

## Materials and methods

This present review was conducted following the Preferred Reporting Items for Systematic Reviews and Meta-Analyses Checklist (PRISMA) guidelines (S1 Appendix) [12].

### Eligibility criteria

The eligibility criteria applied to the studies included/excluded in this review were established according to the PECOS acronym, aiming to address the following focused question: "When compared to healthy individuals, can chronic diseases influence the outcome of endodontic treatment?"

**Population (P).** Included studies involved populations that underwent endodontic treatment in permanent teeth. Studies where endodontic treatment was performed in deciduous

teeth or where endodontic treatment was not performed were excluded. No studies were excluded based on the sex or ethnicity of the population.

**Exposure (E).** Included studies involved populations of interest exposed to pre-existing systemic chronic diseases, as confirmed through validated diagnoses. Chronic diseases were considered those that exhibit one or more of the following characteristics: they are permanent, leave residual disability, are caused by irreversible pathological changes, require special patient training for rehabilitation, or can be expected to necessitate long-term supervision, observation, or care [13]. Studies where the exposure was an acute disease or where individuals were healthy were excluded.

**Comparison (C).** Included studies involved a comparison with healthy individuals. Studies where this condition was not established or those where the comparison was solely with subjects with chronic diseases, even if adequately controlled, were excluded.

**Outcome (O).** Studies were included that assessed the periapical health status using any standardized and validated metric, whether through clinical or radiographic evaluation. Excluded were studies where the outcome of interest was not assessed or where the assessment was not performed using the appropriate method. Studies that only evaluated the survival rate of endodontically treated teeth in the mouth, studies without imaging diagnosis, and studies that assessed the prevalence of periapical changes but did not quantify the number of teeth with a healthy apex were also excluded.

**Study designs (S).** Included study designs encompassed cross-sectional, cohort, case-control, randomized clinical trials, non-randomized trials, or pseudo-randomized trials. Excluded were any descriptive studies, such as editorials, case reports, case series, expert opinions, and guidelines.

### Sources of information and search strategy

Appropriate combinations of keywords and truncations were developed and adapted for the following databases: the Latin American and Caribbean Center on Health Sciences (LILACS), Cochrane Library, Embase, CINAHL, PubMed/Medline, Scopus, and Web of Science (S2 Appendix). Grey literature searches were also conducted through Google Scholar and ProQuest. The searches were initially conducted on November 8, 2021, and updated on October 25, 2023. Manual searches of the references of included articles were performed, and an expert on the team who did not participate in the reading phase was consulted to suggest any relevant publications on the topic. All references were managed using appropriate reference management software (EndNote® Web—Thomson Reuters, Philadelphia, PA), and duplicates were removed.

### Selection process

The article selection process occurred in two phases: Phase 1: Two independent reviewers (B. M.M.A and T.K) read the titles and abstracts of the references retrieved by the search strategy in the databases. All articles that did not meet the eligibility criteria were excluded at this stage. The articles selected in phase 1 were read in full by the same reviewers, and the same eligibility criteria were applied (phase 2). All readings were conducted using the Rayyan website (https:// rayyan.qcri.org), enabling the blinding of the reviewers in all the assessments. In case of disagreement between the two reviewers that could not be resolved through discussion, a third team member (C.M.A) acted as a moderator, providing the tie-breaking vote.

To ensure calibration between the two reviewers, the Kappa coefficient was calculated, and the reading began only when the agreement value was > 0.7, indicating a good agreement.

## Data collection process

The two reviewers (B.M.M.A and T.K) extracted relevant information from the included articles, such as study characteristics (author, publication year, country of origin, study design), sample characteristics (sample size, existing systemic disease, gender, average age), main results, and conclusion.

Similarly, when there was a disagreement that could not be resolved through discussion and mutual agreement between the reviewers, a third reviewer (C.M.A.) acted as a moderator to make the final decision.

If any data of interest to the research were missing or incomplete, two attempts were made to contact the first and last authors of the article to obtain unpublished information. Two email attempts were made, with a one-week interval between them. When there was no response, the article was excluded with a proper justification.

## Data items

Data regarding the frequency of the event (outcome of success or failure of the endodontic treatment) were extracted from the included studies, along with the total sample size for both comparison groups. When the data were available only in graphical format, the web app Webplot Digitizer (https://apps.automeris.io/wpd) was used for data extraction.

## Study risk of bias assessment

The risk of bias assessment was carried out using appropriate tools according to the epidemiological design of each study. For observational studies, the Joanna Briggs Institute tool was used for each type of study (cross-sectional studies and cohort studies). The risk of bias assessment was independently conducted by two reviewers who judged the included articles, marking each assessment criterion as "yes," "no," "unclear," or "not applicable." The risk of bias was classified as high when the study obtained 49% "Yes" scores, moderate when the study reached 50% to 69% "Yes," and low when the study achieved more than 70% "Yes" scores.

For the assessment of randomized clinical trials, the "Cochrane Collaboration tool for assessing risk of bias" was used. This tool evaluates seven different domains: random sequence generation, allocation concealment, participant and personnel masking, outcome assessor masking, incomplete outcome data, selective reporting of results, and other sources of bias. Each assessed domain was judged for possible risk of bias and classified as "high risk" or "low risk" of bias.

Non-randomized clinical studies were assessed using the ROBINS-I tool, which also evaluates seven domains: bias due to confounding, bias in the selection of participants in the study, bias in the classification of interventions, bias due to deviations from intended interventions, bias due to missing data, bias in the measurement of outcomes, and bias in the selection of the reported result. Each domain was classified as low, moderate, or serious risk of bias.

When there was insufficient data in the study that prevented a proper judgment, the risk of bias was considered "unclear." A third reviewer (C.M.A.) acted to resolve any disagreements that persisted even after a consensus meeting between the two reviewers. The robvis web app (https://www.riskofbias.info/welcome/robvis-visualization-tool) was used to generate figures.

## Effect measures

The outcomes assessed in the included articles were reported as binary data. Therefore, the odds ratio (OR) was used as the effect measure to assess the association between individuals with systemic diseases and healthy individuals.

## Synthesis methods

A random-effects meta-analysis was conducted using the statistical software RStudio version 1.2.1335 (Rstudio Inc, Boston, USA), with study weights determined by the Mantel-Haenszel method. Heterogeneity was calculated using the inconsistency index ($I^2$), and variance was estimated using the DerSimonian-Laird method. 95% of confidence intervals (CI) were generated, and the significance level was set at 5%. Separate analyses were performed for each type of systemic disease. A minimum of three articles with the necessary data that met the eligibility criteria for quantitative synthesis was established for each systemic disease.

Articles that were not included in the meta-analysis were presented graphically using a bubble chart for categorical variables. For this, Python programming language was used, along with Matplotlib and NumPy libraries for data manipulation and graphical visualization. Each study was represented by a bubble, and positioned on the x and y axes to indicate the evaluated disease and the article's name, respectively. The color of the bubbles was used to differentiate treatment outcomes, while the size of the bubbles indicated the sample size of patients with the disease. The chart provided a clear and comparative visual representation of the results of the studies included in the systematic review.

## Reporting bias assessment

A publication bias assessment using a funnel plot and Egger's test was planned. However, due to the limited number of available studies (n < 10), it was not possible to carry out this approach. To minimize the possibility of publication bias, a comprehensive search was conducted across various databases, including grey literature, as well as the LILACS database, which covers publications in languages other than English.

## Certainty assessment

The level of certainty of the generated evidence was assessed using the Grading of Recommendations, Assessment, Development and Evaluation (GRADE) tool [14]. This tool classified the generated evidence into four levels of certainty: very low, low, moderate, and high, considering the following domains of evaluation: study limitations, inconsistency, indirect evidence, publication bias, imprecision, risk of bias, dose-response effect, and plausible confounding.

# Results

## Study selection

A total of 2,817 articles were retrieved through the search strategy in the seven electronic databases, resulting in 1,731 references after duplicates were removed. After the analysis of titles and abstracts (phase 1), 39 articles were selected for full-text reading (phase 2). Following the full-text reading of the articles in phase 2, twenty articles were excluded (S3 Appendix). Four studies were included in the update search, resulting in 23 articles included for qualitative synthesis (Fig 1). No articles from grey literature were included, and none were found through manual reference searches or expert consultation.

## Study characteristics

Among the included articles, all were written in English and originated from countries such as Brazil, Croatia, Spain, Finland, India, Iraq, Italy, Jordan, Portugal, the United Kingdom, Sweden, Turkey, and the United States. These articles were published between 1989 and 2021. The sample sizes of the studies ranged from 46 to 5494 participants, with average ages ranging from 18 to 84 years. The chronic diseases assessed included rheumatoid arthritis, diabetes,

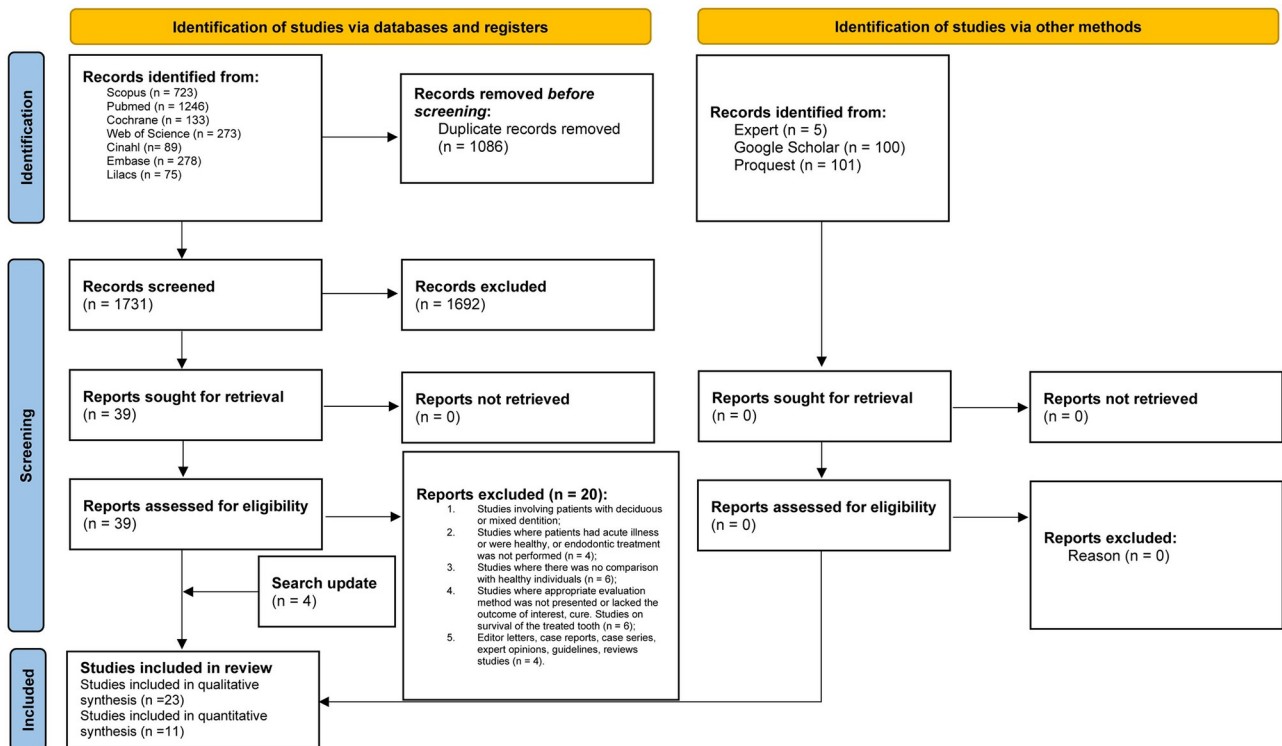

**Fig 1. Flowchart of literature search and selection criteria.** *From*: Page MJ, McKenzie JE, Bossuyt PM, Boutron I, Hoffmann TC, Mulrow CD, et al. The PRISMA 2020 statement: an updated guideline for reporting systematic reviews. BMJ 2021;372:n71. doi: 10.1136/bmj.n71. For more information, visit: http://www.prisma-statement.org/.

HIV, and heart diseases. The studies addressed disease diagnosis, post-treatment follow-up for up to 6 years, and the evaluation of lesion healing or repair. Detailed characteristics of the included studies can be found in Table 1.

## Risk of bias in studies

Regarding the risk of bias in individual studies, five studies were identified as having a high risk of bias, and five studies had a moderate risk of bias (Fig 2). Most of these studies did not adequately control potential confounding factors, such as age, gender, type of endodontic treatment, post-treatment, and the variety of materials used.

## Results of individual studies

The individual results of the included studies were categorized based on the underlying disease assessed.

**Diabetes.**   Healthy patients exhibited a superior periapical condition, albeit without statistical significance [15, 16]. However, endodontic treatment proved effective in preserving treated teeth, even in type 2 diabetic patients with inadequate metabolic control. Periradicular lesions showed a slower healing and repair process in the diabetes group compared to the control group. Diabetic patients displayed a negative response after one year of follow-up. Nevertheless, in both groups, there was repair of the lesions, although the required time differed between them [17, 18]. A significantly higher healing rate was observed in the group of healthy patients compared to the diabetic group, but this disparity was evident only in the 6-month

**Table 1. Characteristics of the included studies.**

| Author, Year (Country) | Study design | Sample (Gender) | Age (Range or Mean and Standard deviation) | Underlying disease | Post-treatment assessment time | Outcomes | Conclusion |
|---|---|---|---|---|---|---|---|
| Alley et al., 2008 (USA) | Cross-sectional | n = 62 (HIV-positive group = 50 teeth / Control group = 50 teeth) | 20–56 | HIV | 3y | There was no statistically significant difference in endodontic success between the two groups. | Dentists can expect endodontic therapy to be as successful for HIV-positive patients as for uninfected patients. |
| Arya et al., 2017 (India) | Cohort | n = 46 (DM Group– 21 / Control group– 25) | DM group: 30–66 / Control group: 30–52 | Type II DM | 3, 6, 9, and 12 months | Both the groups depicted a significant reduction in PAI scores at each follow-up (P < .05). No significant difference in the PAI score was found at baseline or the follow-up time periods in both groups except at the 12-month follow-up (P < .05). In the diabetic group, 43% of the teeth were considered healed (PAI#2) at 12 months, 90% were improved (lower PAI score), and 10% were unchanged (same PAIscore), but none of them showed a worse score. | The most important finding of our study despite the increase in HbA1c levels was the improvement of PAI scores (90% and 100% in diabetic and nondiabetic patients, respectively) after endodontic treatment. These findings suggest that root canal treatment is effective in retaining teeth even in poorly controlled type 2 diabetic patients |
| Aydin et al., 2021 (Turkey) | Case-control | n = 78 (DM group– 46 teeths / Control group– 52 teeths) (34 M, 43 F) | DM group: 38 ± 6,5 / Control group: 43 ± 4,7 | Type II DM | 1y | In both groups, FD values were increased significantly 1-year post-treatment as compared with those prior to treatment (p<0.05). The time-dependent increase in FD was significantly greater in the control group (p<0.05). There was a significant decrease in PAI scores in both type 2 DM and control groups depending on time (p<0.05). No significant difference was found between the groups in terms of time-related decreasing in PAI scores (p>0.05). | An treatment in the fractal dimension of the periapical lesion area was observed 1-year after root canal treatment. Diabetes Mellitus had a negative effect on fractal dimension increase. |
| Britto et al., 2003 (USA) | Cross-sectional | n = 53 (DM group = 30 patients / Control group = 23 patients)(26 M, 27 F) | 39–84 | Type I and type II diabetes | ND | There were significant interactions between sex and diabetes diagnosis for both of the endodontic outcomes, nonsurgical endodontic treatment with lesions and nonsurgical endodontic treatment without lesions. Men with type 2 diabetes who had endodontic treatments were more likely to have residual lesions after treatment. | Type 2 diabetes is associated with an increased risk of ill response by the periradicular tissues toodontogenic pathogens. |
| Cooper et al., 1993 (United Kingdom) | Cross-sectional | n = 48 (HIV group– 40 teeths / control group– 17 teeths) (47 M, 1 F) | 24–68 | HIV | 1 to 3 months | There were no short-term failures of root canal treatment in either group. Both groups showed 100%short-term success of root canal treatment. | Asymptomatic and symptomatic patients with HIV indicates a conventional approach to root canal treatment is acceptable and antibiotic prophylaxis is not required |

*(Continued)*

**Table 1.** (Continued)

| Author, Year (Country) | Study design | Sample (Gender) | Age (Range or Mean and Standard deviation) | Underlying disease | Post-treatment assessment time | Outcomes | Conclusion |
|---|---|---|---|---|---|---|---|
| Falk et al., 1989 (Sweden) | Cross-sectional | n = 266 (Long duration diabetics = 94 / short duration diabetics = 86 / non-diabetics = 86) | 20–70 | Type I DM | ND | Women with long diabetes duration exhibited a higher percentage of endodontically treated teeth with such lesions than women with short diabetes duration or women without diabetes. In long and short duration diabetics and non-diabetics, 39%, 58% and 49%, respectively, had no periapical or juxtaradicular lesions. | Diabetes mellitus does not therefore appear to be of vital importance for periapical infections. A closer analysis, however, revealed that women with long diabetes duration exhibited more endodontically treated teeth with periapical lesions than women with short diabetes duration and women without diabetes. |
| Ferreira et al., 2014 (Portugal) | Cross-sectional | n = 46 (DM Group = 23 / Control Group = 23) | DM group: 64.42 ± 12.97 / Control group: 50.39 ± 12.4 | Type I and type II diabetes | ND | For the analyzed parameters related to the diagnosis pulp, mobility, fistula, pain on percussion horizontal and vertical evaluation of final restoration and the time interval between the query and the final restoration shutter and / or the control visit, there were no differences statistically significant (p > 0.05). Regarding the treatment of the success of endodontic treatment, this was 62% in the test group and 80% in the control group (p > 0.05). | The results of this study are inconclusive regarding the increasing prevalence of apical periodontitis in diabetic patients. Regarding the evaluation of the success of endodontic treatments examined it was found that the success rate in diabetic patients is lower, though not statistically significant. For this reason and given the limitations of this study, we cannot state that patients with diabetes mellitus have a greater predisposition to the development of periradicular lesions or that the success of endodontic treatment in these patients is compromised. It is important, however, that further studies are developed to characterize the pulp and periradicular changes and to assess the prevalence of apical periodontitis and progression in patients with diabetes mellitus. |
| Fouad et al., 2003 (USA) | Cohort | n = 5494 (IDDM group = 70 / NIDDM = 214 / Control group = 5210) | ND | IDDM or noninsulin-dependent DM | > 2y | There was a trend toward increased symptomatic periradicular disease in patients with diabetes who received insulin, as well as flare-ups in all patients with diabetes. Two years or longer postoperatively, 68 percent of cases followed were successful. | Patients with diabetes have a reduced likelihood of success of endodontic treatment in cases with preoperative periradicular lesions. |
| Hussein et al., 2023 (Iraq) | Cross-sectional | n = 72 (DM Group = 36 / Control group = 36) | 25–40 | Type II DM | ND | The ratio of apical periodontitis in root-filled teeth was higher in the DM group (three teeth, 12%) than in the Healthy group (one tooth, 7.1%) | The higher prevalence of apical periodontitis in diabetic patients suggests a positive relationship between worsened apical areas of teeth and health risk factors, such as diabetes mellitus disease and environmental stress. |

(*Continued*)

**Table 1.** (Continued)

| Author, Year (Country) | Study design | Sample (Gender) | Age (Range or Mean and Standard deviation) | Underlying disease | Post-treatment assessment time | Outcomes | Conclusion |
|---|---|---|---|---|---|---|---|
| Ideo et al., 2022 (Italy) | Cohort | n = 188 (AI Group = 99 / Control group = 99) | AI group: 47 ± 13.2 / Control group: 48 ± 15.8 | Auto immune diseases (inflammatory bowel disease; psoriasis; rheumatoid arthritis) | ND | The prevalence of AP was very similar in root canal–treated teeth compared with nontreated teeth in both groups (root canal treated: AI = 52.8% and controls = 58.3%; nontreated: AI = 47.2% and controls = 41.67%). The quality of root canal treatment and coronal restoration of the endodontically treated teeth with AP was similar between groups and was judged adequate in only 15% of cases in the AI group and 10% in the control group. | Understanding the interaction between the host predisposition to inflammatory diseases and the effect of immune modulators on the treatment of AP may help in designing new treatment strategies for AP. Further investigation is needed. |
| Karatas et al., 2020 (Turkey) | Cross-sectional | n = 96 (RA group = 48 / Control group = 48) | (RA group: 47.6 ± 10.6 / Control group: 47.1 ± 10.7) | RA | ND | The prevalence of apical periodontitis in at least 1 tooth was higher in the RA group (47.9%) than in the control (29.7%) (OR = 3.087, P = 0.027). At least 1 root canal treated tooth with apical periodontitis was found in 5 (10.4%) and 6 (12.5%) of the RA and the apical periodontitis patients, respectively. The difference between the groups in terms of the prevalence of one or more root canal treated teeth with apical periodontitis was not statistically significant (P>0.05). | Patients with RA can be more prone to develop apical periodontitis. However, RA did not affect the response to root canal treatment because there was no significant difference between the RA and control groups in terms of root canal treatment teeth with apical periodontitis. |
| Laukkanen et al., 2019 (Finland) | Cross-sectional | n = 504 (DM– 41 teeths / Other immunosuppression– 42 teeths / Cardiovascular disease– 132 teeths / Other systemic diseases 141 teeths/No systemic diseases– 284 teeths) | 51.6 ± 15 | Systemic diseases (Diabetes mellitus, other immunosuppression, cardiovascular diseases and other systemic diseases) | 6–71 months | In the primary analyses, the success rate of root canal treatment was 73.2% in DM patients and 85.6% in patients with no systemic disease (p = 0.043); other systemic diseases had no impact on success. | DM diminished the success of RCT, especially in teeth with apical periodontitis. However, tooth-based factors had a more profound impact on the outcome of RCT. This should be considered in clinical decision-making and in assessment of root canal treatment prognosis. |
| Limeira et al., 2020 (Brazil) | Cross-sectional | n = 150 (DM group = 50 / Control group = 100)(69 M, 81 F) | (DM group: 27.9 ± 6.6 / Control group: 27.9 ± 6.5) | Type I DM | ND | The prevalence of apical periodontitis was 58% in DM participants and 15% in nondiabetic participants (P = 0.00). In the DM group, 52% of the participants presented at least 1 root canal treatment with apical periodontitis, whereas in the nondiabetic group only 8% of the participants exhibited at least 1 root canal treatment with apical periodontitis (P = 0.00). | Root canal treatment, apical periodontitis, and root canal treatment with apical periodontitis were more prevalent in individuals with DM than in nondiabetic individuals. Root canal treatment and apical periodontitis were associated with the presence of DM, specifically root canal treatment with diagnostic time and apical periodontitis with glycemic control. |

(Continued)

**Table 1.** (Continued)

| Author, Year (Country) | Study design | Sample (Gender) | Age (Range or Mean and Standard deviation) | Underlying disease | Post-treatment assessment time | Outcomes | Conclusion |
|---|---|---|---|---|---|---|---|
| López-López et al., 2011 (Spain) | Cross-sectional | n = 100 (DM group = 50 / Control group = 50) | 60.7 ± 10.3 | Type II DM | ND | Diabetic patients with root-filled teeth, 16 (46%) had apical periodontitis affecting at least one treated tooth. Among controls with root-filled teeth, six (24%) had apical periodontitis affecting at least one treated tooth | The data reported in the present study, together with the results of studies conducted so far, are not conclusive but show some differences in the natural history of periapical lesions in the diabetic patient suggesting an association between DM and apical periodontitis. |
| Maniglia et al., 2014 (Brazil) | Cross-sectional | n = 80 (DM group = 40 / Control group = 40)(29 M, 51 F) | 35–70 | Type II DM | ND | At least one tooth was found with apical periodontitis in 90% (n = 32) of the diabetic patients and in 52% (n = 21) of nondiabetic subjects (p = 0.0001). Regarding root-filled teeth, 44% (n = 51) presented apical periodontitis amongst the diabetic patients, whereas only 17% (n = 17) (p = 0.0004) were affected in the control group. The diabetic patients presented larger quantity of apical periodontitis than did the nondiabetics (p = 0.0189). | Type 2 Diabetes Mellitus is associated with an increase in the prevalence of apical periodontitis. |
| Marotta et al., 2012 (Brazil) | Cross-sectional | n = 90 (DM group = 30 / Control group = 60) | 58.2 ± 8 | Type II DM | ND | Of the root canal–treated teeth from diabetic individuals, 46% were associated with apical periodontitis, whereas 38% from nondiabetics evinced disease. However, this difference was nonsignificant (p >.05). | These findings suggest that diabetes can serve as a disease modifier of apical periodontitis in the sense that individuals with diabetes may be more prone to develop primary apical periodontitis. However, our findings do not confirm that diabetes may influence the response to root canal treatment because treated teeth from diabetics had no significantly increased prevalence of disease when compared with controls. |
| Paljevic, et al., 2023 (Croatia) | Cohort | n = 52 (DM group = 26 /Control group = 26) | DM group = 64.5 (53.75–70.25) / Control group = 63.5 (53–71) | Type II DM | 6 and 12 months | Analysis of the results revealed a significantly higher healing rate in the control group compared to the diabetic group only at the 6-month follow-up (66.6 vs. 33.3%; p = 0.0275). Analysis of the full scale PAI index disclosed significantly higher PAI values in the diabetic subjects at 6- and 12-month follow-up. | Root canal treatment remains an effective means of conservative treatment in diabetic patients. While the healing is not compromised, regular follow-ups are necessary to monitor the healing process. |

*(Continued)*

**Table 1.** (Continued)

| Author, Year (Country) | Study design | Sample (Gender) | Age (Range or Mean and Standard deviation) | Underlying disease | Post-treatment assessment time | Outcomes | Conclusion |
|---|---|---|---|---|---|---|---|
| Quesnell et al, 2005 (USA) | Cohort | n = 66 (HIV group = 33 / Control group = 33) | 18–60 | HIV | 1y | Patients in the HIV positive group had significantly lower PAI scores after 1 year than when they started treatment (p < 0.01). Likewise, patients in the control group also showed significantly lower scores at the 1-year follow-up examination (p < 0.01). Both groups also had 29 out of 33 patients (87.9%) show some improvement in score over the study period, with the remaining four either staying the same or getting worse. There were no statistically significant differences between the two with respect to the degree of periradicular healing. | The results indicate that clinicians do not have to alter their expectations for healing and resolution of periradicular lesions based solely on the HIV status of their patients |
| Rudranaik et al, 2016 (India) | Cohort | n = 80 (DM group = 40 / Control group = 40) | 20–60 | Type II DM | 1 week, 2 week, 1 month, 2 month, 6 month and after one year | DM group subjects had chronic and exacerbating lesions with significantly larger lesions (p = 0.029). The clinical evaluation of periapical healing outcome using Strindberg criteria showed 100% success in group I within 1 month, however it was observed after 2 months in Group II. DM group showed 85% success in one year on radiographic evaluation. Poor controlled diabetics showed failure compared to fair and good controlled. | The diabetic patients were more prone for chronic periapical disease with larger lesions. The periapical lesions were more prevalent in diabetic patients than in non diabetics. Smaller sized lesions healed faster where as larger sized lesions showed higher failure rate. Healing outcome at the end of one year was poor in poor controlled diabetics when compared to fair and good controlled diabetics in DM group. The clinical and radiographic healing outcome of single visit endodontic therapy was delayed in diabetic patients. Although few lesions still persisted over a period of one year radiographically, long term follow up would probably have shown further decrease in lesion size, adding to the success rate. |
| Segura-Egea et al, 2005 (Spain) | Cohort | n = 70 (DM group = 38 / Control group = 32) | 63.1 ± 8.3 | Type II DM | ND | Amongst diabetic patients, 10 root filled teeth (83%) had apical periodontitis, whereas in the control subjects 12 root filled teeth (60%) had apical periodontitis (p = 0.17). | Type 2 diabetes mellitus is significantly associated with an increased prevalence of apical periodontitis |

*(Continued)*

**Table 1.** (Continued)

| Author, Year (Country) | Study design | Sample (Gender) | Age (Range or Mean and Standard deviation) | Underlying disease | Post-treatment assessment time | Outcomes | Conclusion |
|---|---|---|---|---|---|---|---|
| Sisli et al., 2019 (Turkey) | Cross-sectional | n = 129 (OGC group = 30 / PGC group = 13 / Control group = 86 patients) (44 M, 85 F) | 56.4 ± 10.3 | Type II DM | > 1y | Significant differences between the DM group and the control group were observed (p < 0.05) in terms of apical periodontitis (the frequencies of both PAI ≥1 and PAI ≥3) and the frequency of cardiovascular disease, while there were no significant differences between the DM subgroups (p > 0.05). | The prevalence of apical periodontitis and severe bone destruction in periapical tissues was significantly higher in the DM patients compared with the nondiabetic patients. |
| Strother et al., 2023 (USA) | Cross-sectional | n = 736 (DM Group = 370 / Control group = 366) | DM Group = 29–93 /Control group = 29–98 | Type I and type II diabetes | ND | The presence of periapical lesions in endodontically-treated teeth was higher in control group when compared to DM+, although no differences were observed for lesion size between groups (P >.05). Diseased/failed treatment outcomes were more frequent in the DM + group (P = .0006) | DM was found to increase the frequency of endodontic treatment, meanwhile reducing the frequency of symptomatic pulpal and periapical diagnoses. |
| Smadi, 2017 (Jordan) | Cross-sectional | n = 291 (DM group = 145 / Control group = 146) | ND | Type II DM | ND | Diabetic group had more endodontically treated teeth (ET) compared with nondiabetic group (4.18 vs 1.82% respectively); this difference was statistically significant (p = 0.001) along with higher apical periodontitis/ET ratio (27.7 vs 19.3 respectively) | This survey demonstrates a higher prevalence of apical periodontitis in DM patients compared with nondiabetic group, with na increased prevalence of persistent chronic AP. |

**Subtitle**: AI–Autoimmune disease; AP–Apical periodontitis; DM–Diabetes Melittus; FD–fractal dimension; HIV–human immunodeficiency virus; IDDM–Insulin-dependent diabetes mellitus; NIDDM–Noninsulin-dependent diabetes mellitus; ND–Not described; OGC–Optimal glycemic control; OR–Odds ratio; PAI–periapical index; PGC–Poor glycemic control; RA–Rheumatoid Arthritis; y–year

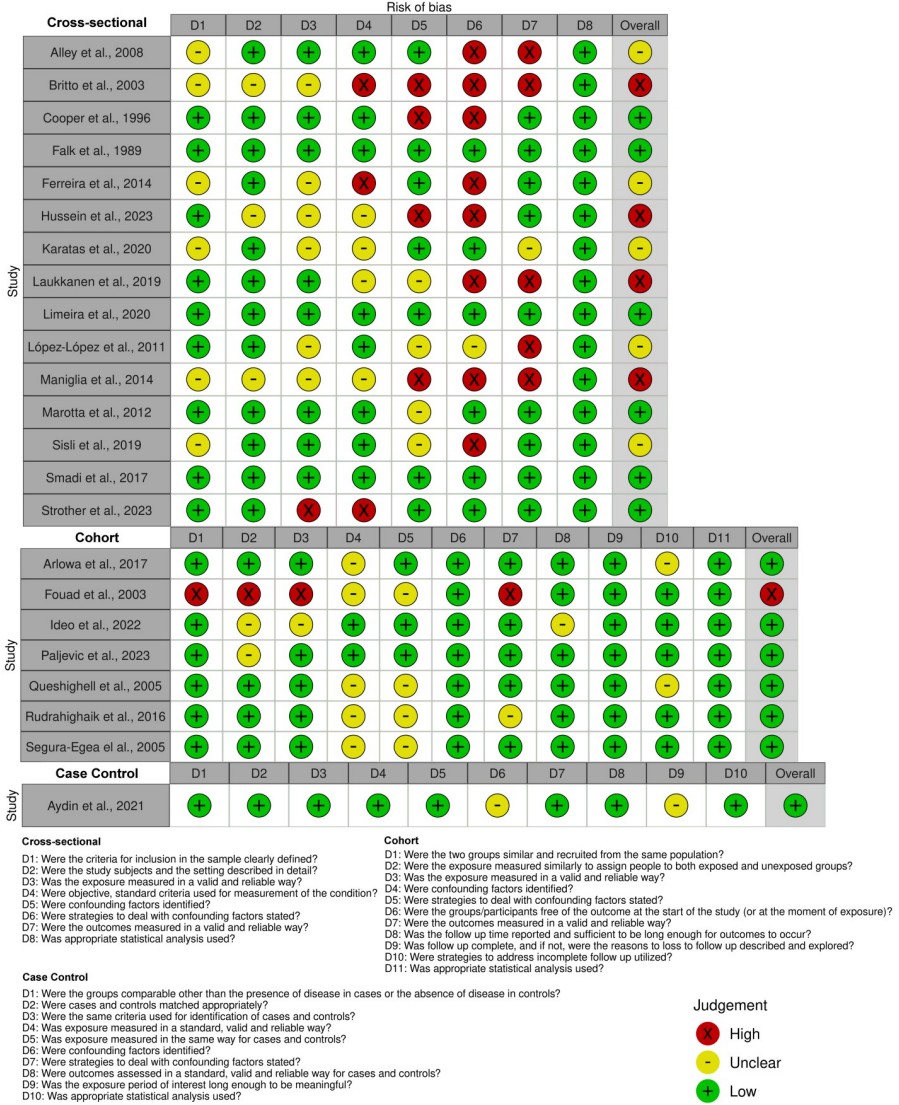

**Fig 2. Risk of bias–Joanna Briggs Institute Critical Assessment Checklist.**

follow-up [19]. There was a slight tendency towards an increase in these lesions in diabetic patients who received insulin. Adequate glycemic control may not reduce the risk of developing these complications in type 2 diabetic patients [20]. After the follow-up period, 68% of the cases showed success in endodontic treatment, although this percentage was reduced in diabetic patients with pre-existing lesions [21]. Additionally, a higher prevalence of apical periodontitis and destruction of periapical and bone tissues was observed in the diabetic group compared to the control group [20, 22–28]. A significant correlation was identified between patients' gender and treatment outcome, concluding that men with type 2 diabetes undergoing endodontic treatment were more likely to have residual lesions after treatment [29]. On the other hand, women with long-term diabetes exhibited a higher proportion of endodontically treated teeth with periapical lesions compared to women with short-term diabetes and women without diabetes [30].

**HIV.**   No statistically significant difference was found in the treatment success between patients with and without HIV, indicating that a differentiated approach is not necessary for individuals infected with the HIV virus undergoing endodontic treatment [31]. In the short-term follow-up period (1 to 3 months) after treatment, no differences in treatment success were observed between patients with or without HIV. The adoption of a conventional approach for HIV patients can be used, dispensing with the use of prophylactic antibiotics [32]. Although an improvement in the condition of apical periodontitis was observed in both HIV and non-HIV patients after treatment and one year of follow-up, there was no statistically significant difference in healing between them. These results suggest that additional specific approaches may not be imperative to achieve success or healing in HIV patients undergoing endodontic treatment [33].

**Autoimmune diseases.**   It was observed that the prevalence of persistent apical periodontitis in at least one tooth was higher in the group of patients with rheumatoid arthritis compared to the control group. However, it is important to note that this difference did not reach statistical significance. These findings suggest a possible association between the presence of the autoimmune disease and the persistence of apical periodontitis, but further studies are needed to confirm this relationship and elucidate possible underlying mechanisms [34]. The prevalence of apical periodontitis was significantly higher in the autoimmune diseases group (65.7%) compared to the control group (46.5%). Furthermore, the study revealed a significant association between smoking and apical periodontitis. Among the autoimmune disease subgroups, patients with rheumatoid arthritis had a lower likelihood of developing apical periodontitis compared to patients with inflammatory bowel disease. The patient's age and the use of tocilizumab were also identified as influential factors in the prevalence of apical periodontitis [35].

**Various chronic diseases.**   A lower success rate in endodontic treatment was observed in patients with diabetes compared to healthy patients. Patients with cardiovascular diseases had relatively good success rates in endodontic treatment, with rates exceeding 80% for teeth with and without preoperative periapical lesions. In the case of other forms of immunosuppression, although success rates were around 78.6%, there was no significant difference compared to healthy individuals. Patients with DM were the only group with a significant difference in the success rate of endodontic treatment compared to patients without systemic diseases. However, further analysis revealed that, in addition to the presence of systemic disease, other dental factors also influenced the success of endodontic treatment. These dental factors were shown to be determinants in the decision-making and prognosis for clinical cases. Therefore, besides the systemic condition, it is essential to carefully consider individual dental factors when planning and performing endodontic treatment in these patients [6].

## Results of the syntheses

The data from eleven studies were subjected to a meta-analysis with the aim of assessing the odds ratio for achieving the periapical index (PAI) scores $\leq$ 2, indicating periapical health when assessed radiographically, in diabetic and non-diabetic patients with teeth that underwent endodontic treatment.

Patients with diabetes were found to have approximately half the odds of achieving PAI scores $\leq$ 2, indicating a lower likelihood of maintaining periapical health compared to non-diabetic patients (OR = 0.46; 95% CI = 0.30–0.70%; $I^2$ = 58%) in teeth that underwent endodontic treatment (Fig 3). These results suggest that diabetes may have a negative impact on periapical health following endodontic therapy.

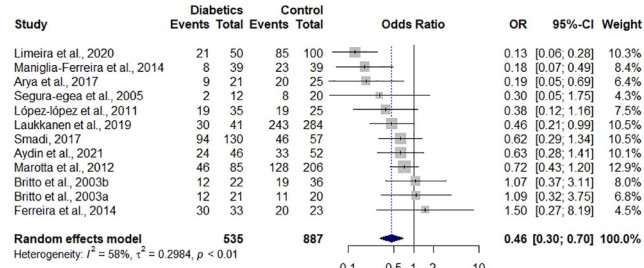

**Fig 3. Forest plot for the assessment of post-endodontic treatment periapical health (PAI ≤ 2), comparing diabetic and non-diabetic patients.**

Regarding the articles that were not included in the meta-analysis, only studies that investigated patients with diabetes showed a difference in the periapical health of teeth treated endodontically compared to healthy individuals. The other chronic diseases did not show significant differences compared to healthy individuals (Fig 4).

## Reporting bias

The funnel plot displayed no asymmetry, a finding supported by the non-statistically significant result from the Egger test (p = 0.728) (Fig 5).

## Certainty of evidence

The certainty of the evidence was classified as very low for the association between diabetes and worsening of periapical health. The decrease in certainty of the evidence occurred due to the existence of only observational studies, the risk of bias present in the included articles, and the inconsistency between the estimates of the included studies, resulting in the presence of heterogeneity (Table 2).

## Discussion

The repair of periapical lesions in endodontically treated teeth depends on the biological response of the host, with tissue healing being influenced by genetic factors and the overall

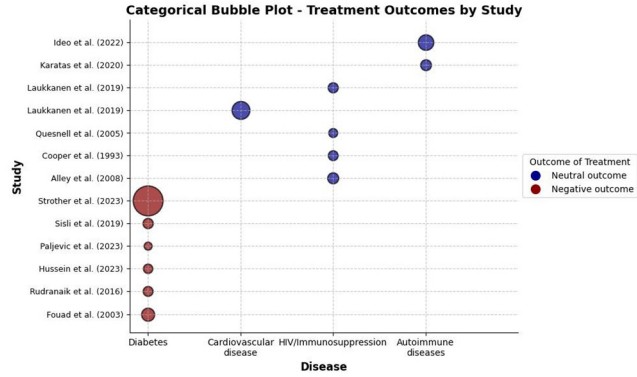

**Fig 4. Bubble plot for categorical variables for the endodontic treatment outcome (negative or neutral) in relation to the underlying disease.**

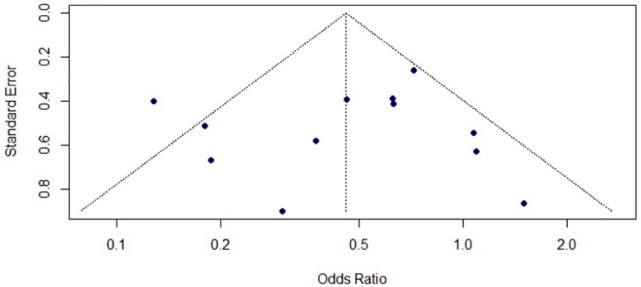

**Fig 5. Funnel plot for assessing the presence of publication bias.**

systemic health of the individual. Therefore, it is crucial for the endodontist to be familiar with systemic factors that can affect the outcome of endodontic treatment, including the effects of systemic diseases on this outcome [36]. The aim of this study was to systematically review the scientific literature on the impact of systemic diseases on periapical health in endodontically treated teeth. Patients with diabetes showed a lower probability of having periapical health compared to healthy patients. Other systemic diseases, such as HIV, cardiovascular disease, and rheumatoid arthritis, did not demonstrate significant differences in relation to the outcome.

Patients with diabetes have a reduced probability of maintaining periapical health following endodontic treatment when compared to healthy patients. These findings are consistent with previous studies that investigated the prevalence of periapical lesions in diabetic patients undergoing endodontic treatment, identifying a higher prevalence of radiolucent periapical lesions in diabetic patients compared to non-diabetic individuals [11, 37]. According to Saghiri et al. (2023), there are discrepancies in the internal structure of dentin in diabetic patients compared to healthy individuals. These discrepancies may have a direct impact on the outcome of endodontic treatment and, consequently, influence the periapical health after endodontic therapy [38]. Despite the presence of moderate heterogeneity in the analysis, in this study, we chose to include only studies that used the Periapical Index (PAI) assessment in the meta-analysis [39]. This approach allowed for the standardization of the assessment method

**Table 2. Summary of findings table.**

**Question: When compared to healthy individuals, can chronic diseases influence the outcome of endodontic treatment?**

| Certainty assessment | | | | | | | of patients | | Effect | | Certainty |
|---|---|---|---|---|---|---|---|---|---|---|---|
| № of studies | Study design | Risk of bias | Inconsistency | Indirectness | Imprecision | Other considerations | Chronic disease | Healthy | Relative (95% CI) | Absolute (95% CI) | |
| Diabetes | | | | | | | | | | | |
| 11 | observational studies | serious[a] | serious[b] | not serious | not serious | none | 307/535 (57.4%) | 655/887 (73.8%) | OR 0.46 (0.30 to 0.70) | 173 fewer per 1.000 (from 280 fewer to 74 fewer) | Very low |

**CI**: confidence interval

**Explanations**

[a]. Presence of studies with moderate and high risk of bias.

[b]. Presence of heterogeneity in the analysis (I-squared = 58%).

across various research studies, considering exclusively the count of teeth that demonstrated periapical health after endodontic therapy (PAI $\leq$ 2).

When considering patients with immunosuppression, whether due to HIV or the administration of immunosuppressive medications as seen in cases of rheumatoid arthritis, no significant difference in the periapical health was observed after undergoing endodontic therapy when compared to healthy individuals. In a study conducted by Laukkanen et al. (2019), which assessed patients with autoimmune diseases, cancer, or those using immunosuppressive medications, no significant difference was found in comparison to healthy individuals. Only three studies addressing the influence of HIV on periapical health after endodontic therapy were identified, and there are no recent studies on this topic. Regarding this specific disease, three articles were included in the analysis. All of them concluded that professionals did not need to adjust their approaches, as there was no difference in the healing and repair process of lesions when compared to groups without the disease. Both groups demonstrated a high success rate after the observation period. Additionally, the use of antibiotic prophylaxis was not shown to be necessary, as it did not exhibit significant effectiveness.

The relationship between cardiovascular diseases and periapical health is still a subject of controversy. According to Jakovljevic et al. (2020), there is a weak association between the presence of cardiovascular disease and periapical disease [40]. On the other hand, in their investigation of the microbiota of periapical periodontitis, Minty et al. (2023) noted a correlation between hypertension and the increased severity of periapical lesions (higher scores on the PAI index) [41]. In this review, only one study examined the impact of cardiovascular diseases on periapical health after endodontic therapy and did not find a significant association with periapical health [6]. It is essential to conduct further research to investigate specific groups of cardiovascular diseases to better understand their relationship with periapical health after endodontic treatment.

It is important to highlight some limitations of the present study. The restriction to include only studies that used a standardized metric to assess the periapical health limited the number of eligible studies. Additionally, the included studies exhibited a moderate to high risk of bias, which reduces the certainty of the analyzed evidence. However, the study addresses a clinically relevant aspect since many patients seeking endodontic treatment may have systemic comorbidities. In the case of diabetic patients, it is crucial to inform them about the possibility of compromised periapical health compared to healthy patients. This underscores the importance of effective communication between healthcare professionals and patients to provide a well-informed and appropriate treatment plan. For future research, it is recommended to conduct studies with rigorous methodologies, incorporating a more comprehensive identification and control of confounding factors. This will enhance the understanding of the impact of various systemic diseases on the periapical health of endodontically treated teeth.

## Conclusion

Diabetic patients showed a lower likelihood of maintaining periapical health. Conversely, patients with HIV, cardiovascular disease, and rheumatoid arthritis did not exhibit significant differences, although the existing evidence is still considered limited. The intricate interplay between systemic health and periapical health underscores the importance of a multidisciplinary approach, especially for diabetic patients. Despite the lack of robust evidence supporting the impact of immunocompromised states and cardiovascular diseases on periapical health, it is crucial to manage these patients in a multidisciplinary manner to provide appropriate care for this population.

## Supporting information

**S1 Appendix. PRISMA 2020 checklist.**
(DOCX)

**S2 Appendix. Database search strategy.**
(DOCX)

**S3 Appendix. Excluded articles (n = 20).**
(DOCX)

## Author Contributions

**Conceptualization:** Bianca Marques de Mattos de Araujo, Odilon Guariza-Filho, Cristiano Miranda de Araujo, Ulisses Xavier da Silva-Neto.

**Data curation:** Bianca Marques de Mattos de Araujo, Erika Calvano Kuchler.

**Formal analysis:** Bianca Marques de Mattos de Araujo, Bruna Marlene de Miranda, Flávio Magno Gonçalves, Erika Calvano Kuchler, Cristiano Miranda de Araujo.

**Funding acquisition:** Everdan Carneiro.

**Investigation:** Bianca Marques de Mattos de Araujo, Tatiana Carvalho Kowaltschuk.

**Methodology:** Bruna Marlene de Miranda, Tatiana Carvalho Kowaltschuk, Flávio Magno Gonçalves, Angela Graciela Deliga Schroder.

**Project administration:** Everdan Carneiro.

**Resources:** Cristiano Miranda de Araujo.

**Software:** Flávio Magno Gonçalves.

**Supervision:** Angela Graciela Deliga Schroder, Odilon Guariza-Filho, Cristiano Miranda de Araujo, Ulisses Xavier da Silva-Neto.

**Writing – original draft:** Bianca Marques de Mattos de Araujo, Bruna Marlene de Miranda, Erika Calvano Kuchler.

**Writing – review & editing:** Bianca Marques de Mattos de Araujo, Tatiana Carvalho Kowaltschuk, Angela Graciela Deliga Schroder, Odilon Guariza-Filho, Everdan Carneiro, Cristiano Miranda de Araujo, Ulisses Xavier da Silva-Neto.

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
