## [Decision Letter · Decision Letter 0]

1 Dec 2023

PONE-D-23-36864IMPACT OF CHRONIC DISEASES ON THE PERIAPICAL HEALTH OF ENDODONTICALLY TREATED TEETH: A SYSTEMATIC REVIEW AND META-ANALYSISPLOS ONE

Dear Dr. de Araujo,

Thank you for submitting your manuscript to PLOS ONE. After careful consideration, we feel that it has merit but does not fully meet PLOS ONE’s publication criteria as it currently stands. Therefore, we invite you to submit a revised version of the manuscript that addresses the points raised during the review process.

We look forward to receiving your revised manuscript.

Kind regards,

Fahad Umer

Academic Editor

PLOS ONE

Journal Requirements:

Reviewers' comments:

Reviewer's Responses to Questions

**Comments to the Author**

1. Is the manuscript technically sound, and do the data support the conclusions?

Reviewer #1: Yes

Reviewer #2: Partly

2. Has the statistical analysis been performed appropriately and rigorously? 

Reviewer #1: Yes

Reviewer #2: Yes

3. Have the authors made all data underlying the findings in their manuscript fully available?

Reviewer #1: Yes

Reviewer #2: Yes

4. Is the manuscript presented in an intelligible fashion and written in standard English?

Reviewer #1: Yes

Reviewer #2: No

5. Review Comments to the Author

Reviewer #1: The article is well written. A thorough research and analyses has been systematically done.

The topic is important and relevant.

The importance of study is properly highlighted and the reasons for conducting it are stated clearly.

The results are presented clearly.

Important findings and results are discussed.

Conclusion is in accordance with data analysis and results.

Overall the Introduction, Methodology and Discussion have covered all aspects related to the study. However shedding a little light on any future recommendations would have made it great.

Reviewer #2: 1. Conclusions are not alligned with the objectives of the study

2. Conclusion is not specific and clear,

3. Inclusion criteria does not mention the inclusion of studies with PAI index only. It generally mentions that the studies with outcome measured with clinical and radiographic outcome were included

4. No data with respect to PAI index in the 23 studies is mentioned, since discussion is revolving around it.

5. The detail under the heading of " results of individual studies" is too lengthy, the information should be crisp and concise and summarized briefly into tabular form.

6. Interprettation of MetaAnalysis specific for diabetic patients is not clearly described with respect to PAI in results.

7. The study is not registered in PROSPERO uptill now.

6. PLOS authors have the option to publish the peer review history of their article (what does this mean?). If published, this will include your full peer review and any attached files.

Reviewer #1: No

Reviewer #2: No

---

## [Author Response · Author response to Decision Letter 0]

4 Dec 2023

Reply to Reviewer #1

The article is well written. A thorough research and analyses has been systematically done.

The topic is important and relevant.

The importance of study is properly highlighted and the reasons for conducting it are stated clearly.

The results are presented clearly.

Important findings and results are discussed.

Conclusion is in accordance with data analysis and results.

Overall the Introduction, Methodology and Discussion have covered all aspects related to the study. However shedding a little light on any future recommendations would have made it great.

ANSWER: We appreciate the comments made and have taken the suggestion into account. Future recommendations have been added to the text in the discussion section.

Reply to Reviewer #2:

1. Conclusions are not alligned with the objectives of the study

ANSWER: We appreciate the feedback and have taken the suggestion into account. Accordingly, the conclusion has been rephrased.

2. Conclusion is not specific and clear

ANSWER: We appreciate the feedback and have taken the suggestion into account. Accordingly, the conclusion has been rephrased.

3. Inclusion criteria does not mention the inclusion of studies with PAI index only. It generally mentions that the studies with outcome measured with clinical and radiographic outcome were included

ANSWER: Thank you for the feedback. We accepted for inclusion any study that used a standardized and validated metric, such as the PAI index. Additionally, other standardized and validated forms were equally considered for inclusion. The decision to employ the PAI index in the meta-analysis was motivated by the fact that the majority of included studies adopted this metric. This clarification has been made in the text.

4. No data with respect to PAI index in the 23 studies is mentioned, since discussion is revolving around it.

ANSWER: Thank you for the feedback. We accepted for inclusion any study that used a standardized and validated metric, such as the PAI index. Additionally, other standardized and validated forms were equally considered for inclusion. The decision to employ the PAI index in the meta-analysis was motivated by the fact that the majority of included studies adopted this metric. This clarification has been made in the text.

5. The detail under the heading of " results of individual studies" is too lengthy, the information should be crisp and concise and summarized briefly into tabular form.

ANSWER: The suggestion was accepted, and the section on individual study results was rewritten. The information for each individual study is provided in Table 1.

6. Interprettation of MetaAnalysis specific for diabetic patients is not clearly described with respect to PAI in results.

ANSWER: Thank you for the feedback. The interpretation of the meta-analysis has been rewritten to make it clearer.

7. The study is not registered in PROSPERO uptill now

ANSWER: We appreciate the comment. The registration was completed at the beginning of the review; however, the verb tense in the text was incorrect. This has been rectified. The protocol number was not included in the text to allow for a blinded peer review.

---

## [Decision Letter · Decision Letter 1]

27 Dec 2023

IMPACT OF CHRONIC DISEASES ON THE PERIAPICAL HEALTH OF ENDODONTICALLY TREATED TEETH: A SYSTEMATIC REVIEW AND META-ANALYSIS

PONE-D-23-36864R1

Dear Dr. Cristiano Miranda de Araujo,

We’re pleased to inform you that your manuscript has been judged scientifically suitable for publication and will be formally accepted for publication once it meets all outstanding technical requirements.

Kind regards,

Fahad Umer

Academic Editor

PLOS ONE

Additional Editor Comments (optional):

Reviewers' comments:

Reviewer's Responses to Questions

**Comments to the Author**

1. If the authors have adequately addressed your comments raised in a previous round of review and you feel that this manuscript is now acceptable for publication, you may indicate that here to bypass the “Comments to the Author” section, enter your conflict of interest statement in the “Confidential to Editor” section, and submit your "Accept" recommendation.

Reviewer #2: All comments have been addressed

2. Is the manuscript technically sound, and do the data support the conclusions?

Reviewer #2: Yes

3. Has the statistical analysis been performed appropriately and rigorously? 

Reviewer #2: Yes

4. Have the authors made all data underlying the findings in their manuscript fully available?

Reviewer #2: No

5. Is the manuscript presented in an intelligible fashion and written in standard English?

Reviewer #2: Yes

6. Review Comments to the Author

Reviewer #2: It is a well written Systematic review and Meta Analysis and all suggestions have been incorporated. One observation is that the Search strategy / or Mesh terms have not been shared in the manuscript.

7. PLOS authors have the option to publish the peer review history of their article (what does this mean?). If published, this will include your full peer review and any attached files.

Reviewer #2: No

---

## [Editor Report · Acceptance letter]

6 Feb 2024

PONE-D-23-36864R1 

PLOS ONE

Dear Dr. de Araujo, 

I'm pleased to inform you that your manuscript has been deemed suitable for publication in PLOS ONE. Congratulations! Your manuscript is now being handed over to our production team.

Kind regards, 

on behalf of

Dr. Fahad Umer 

Academic Editor

PLOS ONE